# Autoinducer Analogs Can Provide Bactericidal Activity to Macrolides in *Pseudomonas aeruginosa* through Antibiotic Tolerance Reduction

**DOI:** 10.3390/antibiotics11010010

**Published:** 2021-12-22

**Authors:** Mizuki Abe, Keiji Murakami, Yuka Hiroshima, Takashi Amoh, Mayu Sebe, Keiko Kataoka, Hideki Fujii

**Affiliations:** 1Department of Oral Microbiology, Institute of Biomedical Sciences, Tokushima University Graduate School, Tokushima 770-8504, Japan; mizukiabu@gmail.com (M.A.); yuka.hiroshima@tokushima-u.ac.jp (Y.H.); hfujii@tokushima-u.ac.jp (H.F.); 2Department of Microbiology and Genetic Analysis, Institute of Biomedical Sciences, Tokushima University Graduate School, Tokushima 770-8503, Japan; kataokakeiko@tokushima-u.ac.jp; 3Department of Clinical Nutrition, Faculty of Health Science and Technology, Kawasaki University of Medical Welfare, Kurashiki 701-0193, Japan; sebe@mw.kawasaki-m.ac.jp; 4Department of Dental Hygiene, Mejiro University College, Tokyo 161-8539, Japan; t.amo@mejiro.ac.jp

**Keywords:** *Pseudomonas aeruginosa*, autoinducer analog, macrolide, antibiotic tolerance

## Abstract

Macrolide antibiotics are used in treating *Pseudomonas aeruginosa* chronic biofilm infections despite their unsatisfactory antibacterial activity, because they display several special activities, such as modulation of the bacterial quorum sensing and immunomodulatory effects on the host. In this study, we investigated the effects of the newly synthesized *P. aeruginosa* quorum-sensing autoinducer analogs (AIA-1, -2) on the activity of azithromycin and clarithromycin against *P. aeruginosa*. In the killing assay of planktonic cells, AIA-1 and -2 enhanced the bactericidal ability of macrolides against *P. aeruginosa* PAO1; however, they did not affect the minimum inhibitory concentrations of macrolides. In addition, AIA-1 and -2 considerably improved the killing activity of azithromycin and clarithromycin in biofilm cells. The results indicated that AIA-1 and -2 could affect antibiotic tolerance. Moreover, the results of hydrocarbon adherence and cell membrane permeability assays suggested that AIA-1 and -2 changed bacterial cell surface hydrophobicity and accelerated the outer membrane permeability of the hydrophobic antibiotics such as azithromycin and clarithromycin. Our study demonstrated that the new combination therapy of macrolides and AIA-1 and -2 may improve the therapeutic efficacy of macrolides in the treatment of chronic *P. aeruginosa* biofilm infections.

## 1. Introduction

*Pseudomonas aeruginosa* is an opportunistic pathogen responsible for chronic infections and biofilm formation, which is a major cause of antibiotic treatment failure [1]. *P. aeruginosa* produces at least three types of the extracellular polymeric substances (EPSs): alginate, Pel, and Psl [2,3]. These EPSs play a critical role in biofilm formation and often cause chronic infections such as cystic fibrosis (CF), diffuse panbronchiolitis (DPB), and chronic sinusitis [4,5].

Macrolides are 14-membered (e.g., erythromycin, clarithromycin), 15-membered (e.g., azithromycin), or 16-membered (e.g., josamycin) antibacterial agents, with an antibacterial spectrum mainly against Gram-positive cocci. Recently, 14- and 15-membered macrolides have attracted attention for their “new actions” in addition to antibacterial activity. This “new action” includes the action of macrolides against biofilm infections such as DPB, despite having almost no antibacterial activity against *P. aeruginosa*, the causative agent of DPB [6,7,8]. It has an immunomodulatory effect on the host and exerts an effect other than antibacterial activity on *P. aeruginosa* [9]. 

In quorum sensing (QS), bacteria sense and respond to their cell density using small diffusible molecules called autoinducers (AI). QS controls several virulence productions [10]. *P. aeruginosa* possesses two signal molecules of *n*-acylated-homoserine lactones (acyl-HSL): *n*-3-oxododecanoyl-homoserine lactone (3-OC12-HSL; OdDHL) and *n*-butanoyl-homoserine lactone (C4-HSL) [11]. We previously demonstrated that the autoinducer 3-OC12-HSL analog-1 (AIA-1) does not inhibit quorum sensing and biofilm formation; however, it enhances the antibacterial activity of biapenem, ofloxacin, and tobramycin in vitro and in vivo [12]. Moreover, it enhances the antibacterial activity against antibiotic-resistant *P. aeruginosa* strains [13].

This study investigated the combination therapy of AIA and macrolides in planktonic and biofilm cells against *P. aeruginosa* PAO1.

## 2. Results

### 2.1. AIA-1 and -2 Do Not Affect Antibiotic Susceptibility

The structures of AIA-1 (*n*-(piperidine-4-yl)-dodecanamide) and AIA-2 (*n*-(pyrrolidine-3-yl)-dodecanamide) are shown in Figure 1a,b. The MICs of antibiotics and AIA-1 and -2 are shown in Table 1. The MICs of AIA-1 and -2 were 128 and 64 μg/mL, respectively, demonstrating unsatisfactory antibacterial activity against *P. aeruginosa*. In the presence of 32 μg/mL AIA-1 and -2, the MICs of the antibiotics did not change, indicating that AIA-1 and -2 did not affect the susceptibility of antibiotics.

### 2.2. Time Killing Assay and MDK_99_ for Azithromycin and Clarithromycin

We examined the effects of the combined use of antibiotics and AIA-1 or -2 on the survival rate of PAO1. The survival rate of *P. aeruginosa* after 10 h was more than 1,000,000 and 10,000 times lower with the combination of azithromycin (256 μg/mL) with AIA-1 (32 μg/mL) and AIA-2 (32 μg/mL), respectively, than with azithromycin (256 μg/mL) alone (Figure 2a). The survival rate of *P. aeruginosa* after 10 h was more than 1000 times and 10,000 lower with the combination of clarithromycin (256 μg/mL) with AIA-1 (32 μg/mL) and AIA-2 (32 μg/mL), respectively, than with clarithromycin (512 μg/mL) alone (Figure 2b). The combination of AIA-1 or -2, with azithromycin or clarithromycin was more effective than azithromycin or clarithromycin alone. 

A new parameter, MDK (the minimum duration for killing a certain percentage of the population), was used to quantify antibiotic tolerance. MDK_99_ represents the minimum duration required for killing 99% of the population [14]. MDK_99_ of azithromycin (256 μg/mL) alone was 8.6 h, and those of the combination of azithromycin (256 μg/mL) with AIA-1 or -2 (32 μg/mL) were both 1.5 h (Figure 2c). MDK_99_ of clarithromycin (512 μg/mL) alone was 15.5 h, and that of the combination of clarithromycin (256 μg/mL) with AIA-1 or -2 (32 μg/mL) was 3 h or 1.8 h, respectively (Figure 2d).

### 2.3. AIA-1 and -2 Affect Azithromycin and Clarithromycin Tolerance in Biofilm Cells

Adding 32 μg/mL AIA-1 and -2 in the killing assay with azithromycin (256 μg/mL) for biofilm resulted in a viable cell number that was approximately 80,000-fold lower than that observed for PAO1 in the presence of azithromycin alone (Figure 3a). Interestingly, the viable cell number of the combined use of even 32 μg/mL azithromycin with AIA-1 and -2 was approximately 100-fold lower than that of azithromycin alone. Similar results were obtained when AIA-1 and -2 (32 μg/mL) were combined with clarithromycin (256 μg/mL), leading to approximately 100-fold lower viable cell numbers compared to when clarithromycin was used alone (Figure 3b). These results suggest that AIA-1 and -2, at a concentration of 32 μg/mL, increased the bactericidal activity of azithromycin and clarithromycin against biofilm cells.

### 2.4. AIA-1 and -2 Affects Bacterial Hydrophobicity of P. aeruginosa

To understand the mechanism of AIA-1 and -2 action on antibiotic tolerance, we measured the hydrophobicity of the cell surface. The adsorption rate of PAO1 to hexadecane was 2.5%, whereas it significantly increased to 18% and 10% when treated with AIA-1 and -2, respectively (Figure 4a).

### 2.5. AIA-1 and -2 Treatment Increases ANS Uptake

To measure the change in cell surface hydrophobicity by AIA-1 and -2, we measured the uptake of the hydrophobic fluorescent probe, ANS. AIA-1 treatment significantly increased ANS fluorescence by approximately 60%, and AIA-2 treatment increased ANS fluorescence by 40% (Figure 4b).

## 3. Discussion

The results of this study show that AIA-1 and -2 improve the bactericidal activity of azithromycin and clarithromycin by changing the cell surface hydrophobicity of *P. aeruginosa*, thereby reducing antibiotic tolerance.

Macrolides have been used to treat chronic respiratory infections caused by *P. aeruginosa*; however, their antibacterial activity against *P. aeruginosa* is unsatisfactory [6,7,8]. Macrolides have many important functions, including anti-inflammatory [15], antivirulence [16,17], QS inhibition [18,19], and antibacterial activities. Sub-MICs of azithromycin inhibit the production of several QS-regulated virulence factors by interfering with the synthesis of autoinducers [18,19]. 

We previously reported that AIA-1, which did not inhibit QS, enhanced the antibacterial effects of biapenem in an in vivo mouse acute infection model and in vitro time kill assay, and enhanced the antibacterial effects of other antibiotics such as levofloxacin and tobramycin. These effects were observed not only in planktonic cells, but also in biofilm cells. AIA-1 did not induce changes in the MIC of antibiotics, suggesting that it exerted its effects on antibiotic tolerance without affecting the MIC levels [12]. In this susceptibility test, the MICs for azithromycin and clarithromycin did not change in the presence of AIA-1 and -2. This result also showed that AIA-1 and -2 could affect the antibiotic tolerance and enhance the bactericidal activity of macrolides. 

Antibiotic tolerance is the ability of bacteria to survive but not grow under antibiotic stress without any gene mutations [20], and it is closely related to the clinical symptoms and prognosis of respiratory tract infections caused by *P. aeruginosa* [21]. In the present study, the MDK_99_ of azithromycin or clarithromycin with AIA-1 and -2 was significantly shorter compared to that of azithromycin or clarithromycin alone. Although the killing assay is generally performed to evaluate antibiotic tolerance, MDK has been confirmed to be an excellent parameter for the quantification of antibiotic tolerance.

Macrolides are effective mainly against Gram-positive bacteria, but are not sensitive to most Gram-negative bacteria, except for *Moraxella catarrhalis*, *Haemophilus influenzae*, and *Bacteroides fragilis* [22]. In most Gram-negative bacteria, the cell surface is hydrophilic because of the O-antigen of lipopolysaccharide molecules in the outer membrane. There are no pathways across the outer membrane of Gram-negative bacteria for the entry of hydrophobic antibiotics or macrolides. Our study demonstrated that adding AIA-1 and -2 changed the cell surface hydrophobicity and accelerated the outer membrane permeability of hydrophobic antibiotics such as azithromycin and clarithromycin. 

Bacterial biofilms are more tolerant to antibiotics compared to planktonic bacteria. However, the mechanisms of antibiotic tolerance in biofilm cells are unknown. We previously reported that the *psl* genes, activated by surface adherence through elevated intracellular c-di-GMP levels, confer tolerance to biapenem [23], and carbon metabolism, iron regulation, and stress response play an important role in antibiotic tolerance to biapenem in biofilm cells [24]. Nguyen et al. showed that the antibiotic tolerance of nutrient-limited and biofilm *P. aeruginosa* is mediated by active responses to starvation [25]. In this study, AIA-1 and -2 surprisingly improved the killing activity of azithromycin and clarithromycin even in biofilm cells. It is still unclear whether this effect is only due to the permeability of macrolides or whether AIA-1 and -2 affect other mechanisms of antibiotic tolerance.

Finally, our study demonstrated that the new combination therapy of macrolides with AIA-1 and -2 could improve the therapeutic efficacy of the current macrolide therapy for chronic *P. aeruginosa* biofilm infections.

## 4. Materials and Methods

### 4.1. Bacterial Strain

*P. aeruginosa* PAO1 was used in all the experiments. It was incubated at 37 °C in lysogeny broth (LB) or on lysogeny agar (LA) plates. 

### 4.2. Reagents

Azithromycin was purchased from Meiji Seika Pharma Co., Ltd. (Tokyo, Japan). Clarithromycin was purchased from Taisho Pharmaceutical Holdings Co., Ltd. (Tokyo, Japan).

### 4.3. Susceptibility Testing for Planktonic Bacteria

The minimum inhibitory concentration (MIC) of planktonic bacteria for AIA-1, -2, azithromycin, and clarithromycin were assessed using a standard microbial broth dilution method [26]. In susceptibility assays including both antibiotics and AIA-1 or -2, AIA-1 or -2 were used at a concentration of 32 μg/mL.

### 4.4. Time Kill Assay and Measuring MDK_99_

PAO1 was grown until it displayed an optical density at 600 nm (OD_60__0_) of 0.25 in LB medium. Subsequently, cells were incubated in the presence of azithromycin (256 μg/mL), clarithromycin (512 or 256 μg/mL), and/or AIA-1 or -2 (32 μg/mL) for various periods of time (0 to 24 h). Assuming that survival at time 0 was 100%, the colony forming unit (cfu) values were converted to percentages.

MDK (the minimum duration for killing a certain percentage of the population)_99_ was measured as the minimum duration required to kill 99% of the cells from killing curves [14].

### 4.5. Biofilm Killing Assay

The biofilms were grown using the MBEC physiology and genetics assay kit (Innovotech Inc., Edmonton, AB, Canada) for 24 h at 37 °C with aeration [27]. In the killing assay for biofilm cells, biofilms were exposed to 32 μg/mL AIA-1 or -2 at 37 °C for 1 h. Biofilms were then exposed to 256, 128, 64, and 32 μg/mL azithromycin or 256 μg/mL clarithromycin at 37 °C for 24 h. Subsequently, the biofilms were washed and disrupted by sonication (Branson 3510, Branson Ultrasonics Corp., Brookfield, CT, USA) for 1 h. Viable counts were determined on LA plates, and the number of CFU per peg was calculated.

### 4.6. Hydrocarbon Adherence Assay

PAO1 was grown until it displayed an OD_600_ of 0.25 in LB medium. Cells were incubated in the presence of AIA-1 or -2 (32 μg/mL) for 4 h. Cells were harvested by centrifugation and resuspended in phosphate urea magnesium sulfate buffer (pH 7.1) at a turbidity of 0.5 at 660 nm for the hydrocarbon adherence assay. Cell surface hydrophobicity was estimated using a modification of the hydrocarbon adherence technique described by Rosenberg et al. [28]. Two hundred microliters of hexadecane (Sigma–Aldrich, St. Louis, MO, USA) were added to 3 mL of microbial suspension in a test tube (12 × 105 mm). The test tube was vigorously agitated using a mixer for 60 s and then allowed to stand.

The turbidity of the aqueous phase was measured using a spectrophotometer, and the hydrophobicity was expressed as the percentage reduction in the initial turbidity of the aqueous suspension.

### 4.7. Cell Membrane Permeability Assays

Cell membrane permeability assays was performed essentially as previously described [29,30]. The fluorescent probe, 8-anilino-1-naphthalenesulfonic acid (ANS; Sigma–Aldrich, St. Louis, MO, USA), was used to assess the integrity of bacterial cell membrane. ANS is a neutrally charged, hydrophobic probe that fluoresces weakly in aqueous environments but exhibits enhanced fluorescence in nonpolar/hydrophobic environments. PAO1 was grown until it displayed an OD_600_ of 0.25 in LB medium. Cells were incubated in the presence of AIA-1 or -2 (32 μg/mL) for 4 h. Cells were washed and resuspended in 5 mM sodium 4-(2-hydroxyethyl)piperazine-1-ethanesulfonic acid (HEPES, pH 7.2) and 98 μL of sample were added to 96-well plate. Subsequently, 2 μL of 3 mM ANS were added to each well, and the fluorescence was measured using a spectrophotometer, with fluorescence emission at 510 nm after excitation at 375 nm.

### 4.8. Statistical Analyses

Data are presented as mean ± standard deviation (SD). All comparisons between the populations were performed using Student’s *t*-test or Fisher’s exact test. All statistical analyses were performed using GraphPad PRISM 5.01 (GraphPad Software, Inc., La Jolla, CA, USA). Statistical significance was set at *p* < 0.05.

## Figures and Tables

**Figure 1 antibiotics-11-00010-f001:**
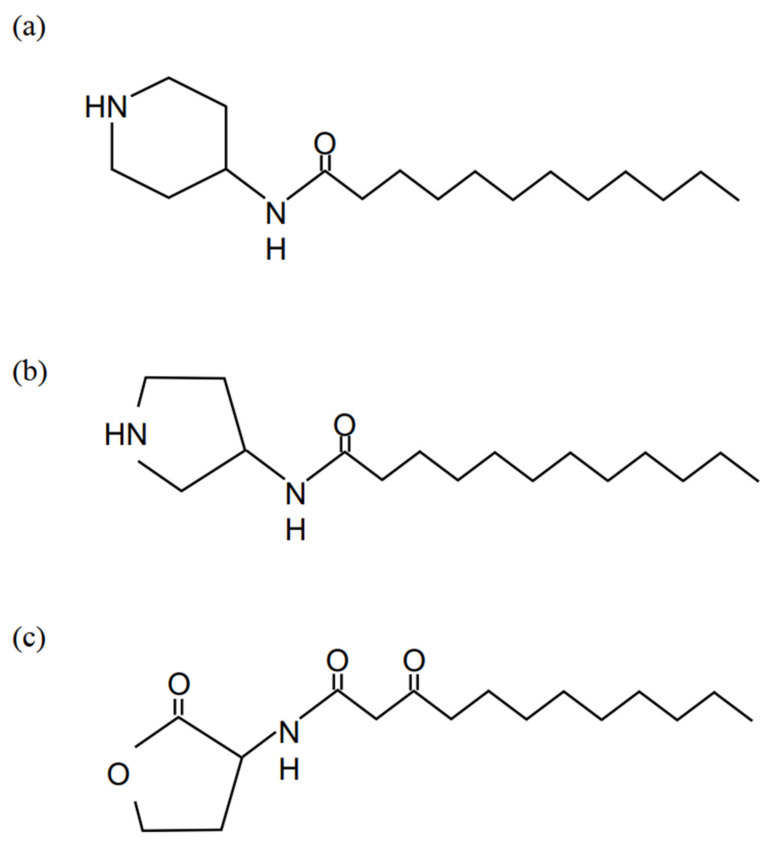
The structure of AIA-1 (*n*-(piperidine-4-yl)-dodecanamide) (**a**), and AIA-2 (*n*-(pyrrolidine-3-yl)-dodecanamide) (**b**), the analog of OdDHL (*n*-3-oxododecanoyl-homoserine lactone). The structure of OdDHL (**c**).

**Figure 2 antibiotics-11-00010-f002:**
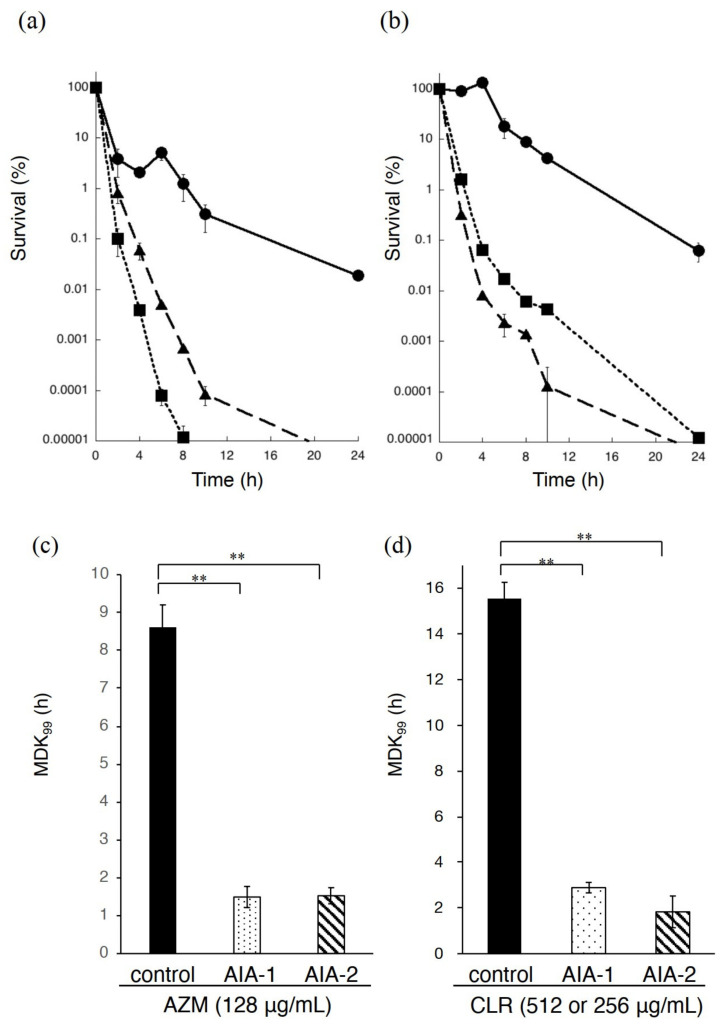
Time-kill assay for PAO1. (**a**) Time-kill assay for PAO1 in the presence of azithromycin at 256 μg/mL with or without AIA-1 and -2 at 32 μg/mL. Azithromycin alone, filled circle on the solid line; azithromycin with AIA-1, a filled square on the dotted line; azithromycin with AIA-2, filled triangle on the dashed line. (**b**) Time-kill assay for PAO1 in the presence of clarithromycin at 512 or 256 μg/mL with or without AIA-1 or -2 at 32 μg/mL. Clarithromycin alone, a filled circle on the solid line; clarithromycin with AIA-1, filled square on the dotted line; clarithromycin with AIA-2, filled triangle on the dashed line. MDK_99_ was determined by measuring the time to kill 99% of the population with azithromycin at 256 μg/mL with or without AIA-1 and -2 at 32 μg/mL (**c**) and clarithromycin at 512 or 256 μg/mL with or without AIA-1 and -2 at 32 μg/mL (**d**). Assuming that survival at time 0 was 100%, CFU values were converted to percentages. All experiments were performed in triplicate. ** *p* < 0.01, (Student’s *t*-test). Error bars, SDs for three experiments.

**Figure 3 antibiotics-11-00010-f003:**
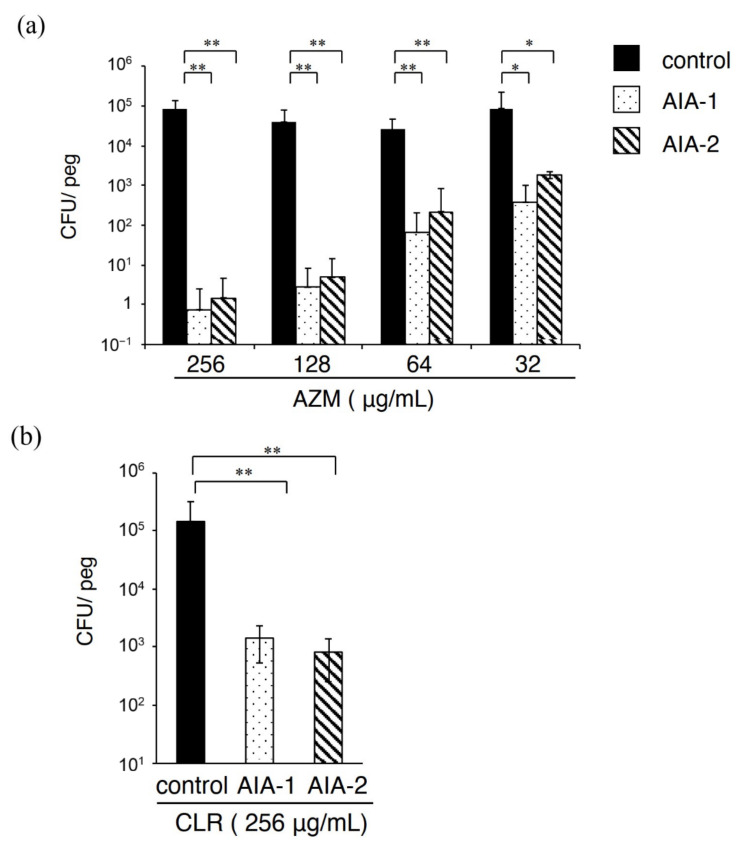
Efficacy of AIA-1 and -2 against PAO1 biofilms. Killing assay for biofilm cells using (**a**) 256, 128, 64, and 32 μg/mL azithromycin with 32 μg/mL AIA-1 or -2 (**b**) 256 μg/mL clarithromycin with 32 μg/mL AIA-1 or -2. The number of CFU per peg was calculated. * *p* < 0.05, ** *p* < 0.01 (Student’s *t*-test). Error bars, SDs for eight experiments.

**Figure 4 antibiotics-11-00010-f004:**
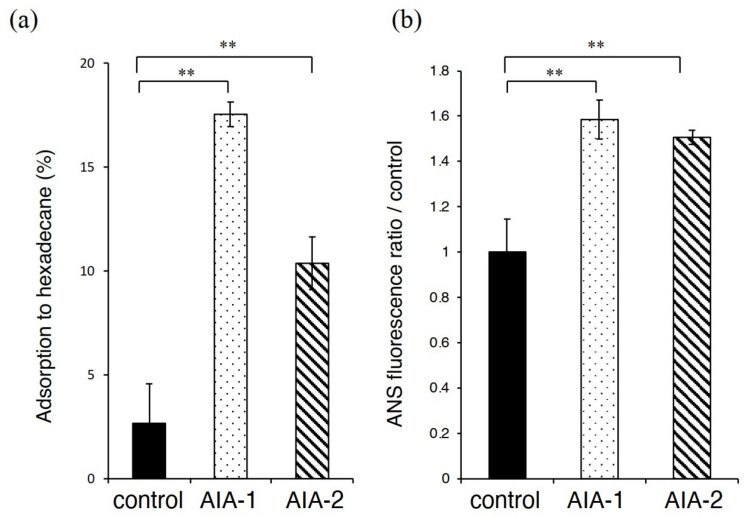
Hydrophobicity of PAO1 treated with AIA-1 or -2. (**a**) The rate of adsorption to hexadecane (cell-surface hydrophobicity) (**b**) ANS (8-anilino-1-naphthylenesulfonic acid) fluorescence ratio by treatment with 32 μg/mL AIA-1 or -2. ** *p* < 0.01 (Student’s *t*-test). Error bars, SDs for four experiments.

**Table 1 antibiotics-11-00010-t001:** Susceptibility to antibiotics and AIAs.

Antibiotics and AIAs	MIC (μg/mL)
AZM	256
CLR	128
AIA-1	128
AIA-2	64
AZM + AIA-1 (32 μg/mL)	256
AZM + AIA-2 (32 μg/mL)	256
CLR + AIA-1 (32 μg/mL)	64
CLR + AIA-2 (32 μg/mL)	64

AZM; azithromycin, CLR; clarithromycin.

## Data Availability

The data presented in this study are available on request from the corresponding author.

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
