# Peer review of "Autoinducer Analogs Can Provide Bactericidal Activity to Macrolides in Pseudomonas aeruginosa through Antibiotic Tolerance Reduction"

_antibiotics, 2021, doi:10.3390/antibiotics11010010_

Round 1

Reviewer 1 Report

The manuscript entitled “Autoinducer Analogs Can Provide Bactericidal Activity to macrolides in Psudomounas aeruginosa Through Antibiotic Tolerance Reduction” by Hideki Fujii and co-workers reports the study of using AIA-1 and AIA-2 as the autoinducer that improves the antibiotic activity of macrolides against chronic P. Aeruginosa biofilm infections. This work is a continuation of their previous work where they found that AIA-1 enhances the antibacterial activity of biapednem, ofloxacin, and tobramycin in vitro and in vivo studies.  

However, several issues need to be addressed: 

1. The description of Figure 2c and Figure 2d is not correctly written or formatted. 

2. There is a “Figure 2” written on the top right of page 4, which needs to be removed. 

3. As described on page 3, lines 78-84, the MDK99 of Azithromycin with 256 ug/ml is 10.7 hr, and MDK of clarithromycin with 512 ug/ml is 12 hr. The data does not match the description of the Figure at all.  

3. In the supplementary material section, there is no information regarding the “Author Contributions” nor ”Funding. “   

4. The format of the “References” is not consistent, which needs significant revisions. 

In summary, I would recommend only after major revision. 

Regards

Author Response

Response to reviewer 1 comments

We would like to thank the editor and reviewers for their helpful comments and hope that we have now produced a more balanced and better account of our work.  We believe that the revised manuscript is acceptable for publication in antibiotics 

Reviewer 1 comments:

1. The description of Figure 2c and Figure 2d is not correctly written or formatted.

We corrected the data in Figure 2c.

2. There is a “Figure 2” written on the top right of page 4, which need to be removed.

We will ask the editorial office in revised version.

3. As described on page 3, line 78-84, the MDK99 of Azithromycin with 256 mg/ml is 10.7 hr, and MDK99 of Clarithromycin with 256 mg/ml is 12 hr. The data does not match the description of the Figure at all.

I rewrote these parts to “MDK99 of azithromycin (256 mg/ml) alone was 8.6 h” and  “MDK99 of clarithromycin (512 mg/ml) alone was 15.5 h”.

4. In the supplementary materials section, there is no information regarding the “Author Contributions” nor “Funding”.

When we submitted this manuscript, we have filled the question columns about author contribution and funding.

5. The format of the “References” is not consist, which needs significant revisions.

We rewrote “References” according to the guideline.

I sincerely appreciate the constructive suggestions made by the reviewers that improved the clarity and message of the manuscript.

Thank you and best wishes,

Keiji Murakami, DDS, Ph.D.

Reviewer 2 Report

There should have been an elaborated background for autoinducer analogs, where the rationale and the hypothesis would have been clearly explained for the work.

Figure 1 should be introduced first, then the result table 1 should come. the caption of figure 1 should be elaborated.

Figure 2, a) and b) should have proper labeling for each trend line.

Overall, it would have been highly appreciated if the combined effect was studied in vivo infection model to check whether it gives consistent results with the previous studies.  Especially, odDHL was shown to have an effect on the host immune system,  I am wondering about the effect of odDHL analogs on the host model.

Author Response

We would like to thank the editor and reviewers for their helpful comments and hope that we have now produced a more balanced and better account of our work.  We believe that the revised manuscript is acceptable for publication in antibiotics 

Reviewer 2 comments:

1. Figure 1 should be introduced first, then the results table 1 should come. The caption of figure 1 should be elaborated.

We will ask the editorial office to change the position of table1 and figure 1.We rewrote the material methods and figure legends about the structures of AIA-1 and -2 to “The structure of AIA-1 (N-(piperidine-4-yl)-dodecanamide) (a), AIA-2 (N-(pyrrolidine-3-yl)-dodecanamide)”. 

2. Figure 2 a, b should have proper labeling for each trend line.

We changed the size of each marker in Figure 2 a, b.

3. It would have been highly appreciated if the combination effect was studied in vivo infection model to check whether it gives consistent results with the previous studies. Especially, OdDHL was shown to have an the host immune system, I am wondering about the effect of OdDHL analogs on the host model.

According to reviewer’s comment, the in vivo mouse infection model was very important tool to confirm these in vitro results. Unfortunately, it is not possible to carry out mouse infection experiment because of the circumstances of our collaborators. We would like to resume it in the future.We previously confirmed the safety in mouse model at this concentration (32 mg/ml) of AIA-1 and AIA-2. We think the effects of AIA-1 and -2 on the host immune system as a next research.

I sincerely appreciate the constructive suggestions made by the reviewers that improved the clarity and message of the manuscript.

Thank you and best wishes,

Keiji Murakami, DDS, Ph.D.

Round 2

Reviewer 1 Report

none